# Give me a hint: Can LLMs take a hint to solve math problems?

**Vansh Agrawal, Pratham Singla, Amitoj Singh Miglani, Shivank Garg, Ayush Mangal**
Vision and Language Group
Indian Institute of Technology, Roorkee
`{vansh_a@ph,pratham_s@me,amitoj_sm@ph,shivank_g@mfs,amangal@cs}.iitr.ac.in`

## Abstract

While state-of-the-art LLMs have shown poor logical and basic mathematical reasoning, recent works try to improve their problem-solving abilities using prompting techniques. We propose giving "hints" to improve the language model's performance on advanced mathematical problems, taking inspiration from how humans approach math pedagogically. We also test robustness to adversarial hints and demonstrate their sensitivity to them. We demonstrate the effectiveness of our approach by evaluating various diverse LLMs, presenting them with a broad set of problems of different difficulties and topics from the MATH dataset and comparing against techniques such as one-shot, few-shot, and chain of thought prompting. Our code is available at https://github.com/vlgiitr/LLM-Math

## 1 Introduction

The ability to reason and logically solve complex mathematical problems is essential for progress in nearly every field, whether it be modeling complex environments, developing new algorithms, or engineering new devices. Recent works have explored Large Language models' mathematical capabilities and found them lacking in logic and basic mathematical reasoning [1] [2]. Improving these reasoning capabilities is important in a future where LLMs can be used to assist humans in increasingly complex mathematical tasks or act like AI math teachers. In this work, taking inspiration from how humans are taught math pedagogically, we propose "hinting" LLMs by giving subtle guidance or clues as a method to improve problem-solving capabilities on mathematical tasks and then compare its effectiveness against other prompting methods, such as one-shot [3], few-shot [3], and chain of thought prompting [4]. We evaluate our approach using the MATH dataset [5], consisting of math problems categorized into distinct classes based on subtopics such as algebra, probability, geometry, etc., with different levels of complexity (1-5). We use a diverse set of LLMs, including base language models, instruction fine-tuned models, models specifically tuned for mathematical tasks, and closed source models such as GPT-4o-mini [6] and Gemini Flash [7] for our evaluation.

We further examine the robustness of these models to adversarial prompts, misleading hints, and clues of varying levels. We investigate how sensitive the models are to incorporating these incorrect hints as context, which may degrade performance, versus their ability to reject such misleading information. Through these experiments, we seek to contribute to the ongoing research on improving the current state-of-the-art language model's reasoning capabilities and their practical applications in solving mathematical tasks [8].

### 1.1 Background and Related Work

Various prompting methods have been shown to increase the accuracy of LLMs in solving complex problems that require understanding and reasoning [9]. A few popular methods being -

38th Conference on Neural Information Processing Systems (NeurIPS 2024).

1. **One shot prompting [3]:** Giving a single example problem and its final answer to the model in-context to learn from.

2. **Few shot prompting [3]:** Providing multiple in-context example problems and their final answers instead of one.

3. **Chain of thought prompting (CoT) [4]:** Providing a detailed step-by-step solution to the in-context example to provide intermediate reasoning steps to solve a problem.

However, these methods have not been extensively explored in the context of solving more complicated mathematical problems. Further, their generalization capabilities, to apply learned knowledge to a broader domain of questions (e.g., algebraic equations, geometry problems) rather than specific problems (e.g., solving a specific algebraic equation, finding roots for a quadratic polynomial), have not been sufficiently researched [10]. Previous math benchmarks[10] [11] show that math is a valuable ability to test an LLM's reasoning and problem-solving capabilities. While there has been work on hinting [12] [13] as a prompting technique, they have not been robust and diverse enough to see if these techniques are generalizable to various types of models and what the effect and sensitivity of these models to adversarial hinting can be.

## 2   Method

We first prompt the Gemini 1.5 flash model to generate hints for our dataset by passing it the question and final answer. These hints are then provided in context to the model and the target problem to help the model reason about the task. We believe this aligns more with how humans solve math problems by getting hints instead of the complete solution for an example problem, as in Chain of Thought, or just the final answer, as in one/few-shot approaches.

To test the adversarial robustness of these models to hints, we provided either an adversarial misleading hint or a hint from a random question to observe how sensitive the models are to our hints. Similar versions for one/few shot and CoT prompting are also generated as shown in Figure 1. More details about the prompting process can be found in Appendix A.

We evaluate a diverse ensemble of LLMs ranging from base models to instruct fine-tuned models and math-finetuned LLMs, including nine open-source models and two closed-source models (More details in Appendix C). We use the MATH dataset [5], which contains problems of seven different classes (e.g., algebra, geometry, etc.) of varying difficulty levels from 1 (easy) to 5 (hard), more details about the difficulty levels are given in Appendix B.

## 3   Experiments and Results

### 3.1   Setup

We evaluate a set of 11 models for our experiments, using open-source implementations where possible and public API offerings for closed-source models. We evaluate 17 different types of prompting techniques, ranging from baseline to hinting to adversarial, along with one/few shot and CoT as given in Appendix A. The problem set consisted of all seven topics from the MATH dataset [5], with 100 problems for each topic. Exact details about the data split are given in Appendix B. We report the comparison by checking the fraction of questions that the model got correct.

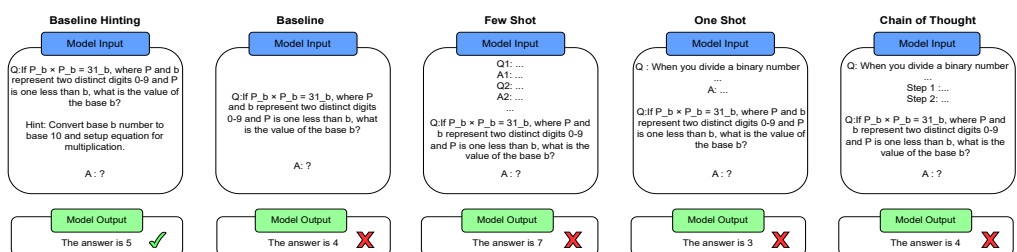

Figure 1: A comparison of various prompting techniques

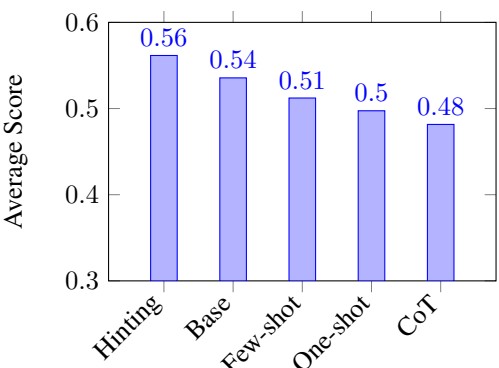 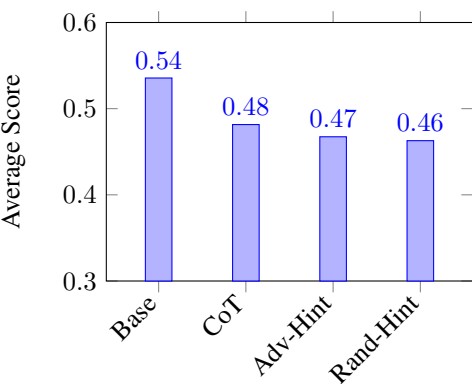

Figure 2: Left: Comparing different prompting techniques, hinting boosts performance and CoT performs worst, as explained in section 3.2. Right: Effect of adversarial hints: Adversarial hinting greatly reduces performances, as explained in section 3.3

## 3.2 Evaluating Hint based prompting

We observe that hinting improves the performance of models, as shown in Figure 2. We attribute the poorer performance of other approaches to a lack of generalization. The improvement of hinting over chain of thought, can be explained as the latter restricts the model's generalization by forcing it to follow the entire reasoning steps of a solution, which might not be the same for all the problems. On the other hand, hints only provide certain helpful directions, giving the model more freedom to generalize better to different problems.

Approaches like the chain of thought [4] give step-by-step solutions, which might not generalize to other problems restricting the search space of these models. This can also be due to a "snowball" effect [5], in which intermediate-generated steps with mistakes can derail the model from the logical direction to the right answer. One-shot and few-shot perform relatively better as they do not restrict the steps, however, the model still gets confined to a narrower subdomain, with few-shot showing more generalisation due to more examples and hence, better performance. Hinting leads to better performance as it helps the model reason about question-specific knowledge more than other prompting techniques and gives initial steps to ensure the right direction without over-fitting to an exact solution.

## 3.3 Evaluating Adversarial Hinting

We find that giving Adversarial hints drastically reduces the model's performance, dropping it below CoT, which performed the worst in our non-adversarial approaches, as shown in Fig 2 and Table 1. We also apply this to few-shot[3] and one-shot[3] settings and find that adversarial hinting affects the performance in those cases as well. The inferior performance of CoT and its prompting variants can be attributed to over-fitting various factors within irrelevant information that influence the model's sensitivity to distractions based on lexical similarity. [14].

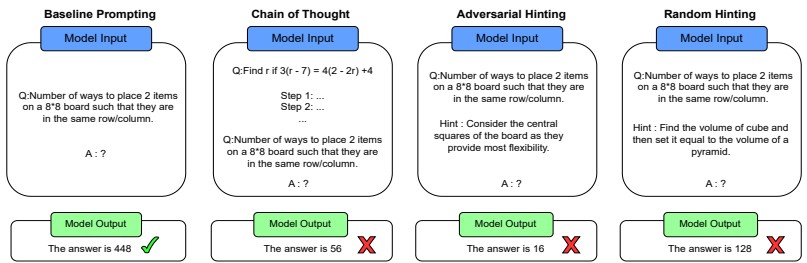

Figure 3: Adversarial and random hinting strategies

| Prompt Technique | Baseline | Few-shot | One-shot | CoT |
|---|---|---|---|---|
| **Baseline** | 0.5355 | 0.5122 | 0.4973 | 0.4815 |
| **with hint** | **0.5615** | 0.5171 | 0.4675 | - |
| **adversarial prompting** | 0.4674 | 0.4968 | 0.4742 | 0.4761 |
| **random prompting** | 0.4628 | 0.4711 | 0.4753 | 0.5073 |

Table 1: Comparison of various prompting techniques with adversarial hinting. Base CoT already entails in-context steps and hence doesn't involve hinting. We observe a drop in performance due to both, adversarial and random hinting, indicating a sensitivity to hinting.

| Model category | Baseline | Hinted | Adv hint | Rand hint |
|---|---|---|---|---|
| **Small models (<=2B)** | 0.3314 | 0.3686 | 0.3143 | 0.2314 |
| **Base models** | 0.5385 | 0.5104 | 0.4898 | 0.2871 |
| **Instruct tuned models** | 0.5471 | 0.5601 | 0.4985 | 0.4971 |
| **Math Fine-tuned models** | 0.6180 | 0.6485 | 0.5581 | 0.5828 |
| **Closed-Source models** | **0.6685** | **0.7186** | **0.6557** | **0.6928** |

Table 2: A comparison of various model types. We find that hinting performs better for all models except the base models, which in general are worse at instruction following. We also notice a drastic drop in performance for all model types when given adversarial hints

## 3.4 Model-wise performance

While comparing the performance of different models, we observe that closed-source models exhibit the highest performance, followed by models fine-tuned for mathematical tasks, instruct-tuned models, and finally, base models, as shown in table 2. Additionally, variations in model size impact performance, with smaller models generally performing worse. However, performance also depends on the extent of fine-tuning; for example, the Qwen-2-Math-Instruct model[15] achieves comparable results to GPT-4o-mini[6] as shown in Appendix E. Among the 7B and 8B models, Qwen-2-Math-Instruct[15] performs best, while Mistral-Instruct[16] ranks the lowest. Our observations further reveal that base models struggle to incorporate and utilize hints effectively, performing worse than the baseline, likely due to their limited ability to follow instructions[17]. We list our exhaustive results in Appendix D.

Further, we also evaluated hinting multimodal models on visual and mathematical tasks and found a similar improvement due to hinting and deterioration due to adversarial hinting. We provide more details in Appendix C.1.

## 4 Conclusion

In this work, we have evaluated the mathematical reasoning abilities of various models and approaches. Our results indicate that providing hints is more effective than giving direct answers because it guides the model to the correct solution instead of restricting its search space like few-shot[3], one-shot[3], and chain of thought[4] which lack generalization. However, CoT may outperform hinting in some cases where the example and target problem are very similar. Hence, for math problems with unknown solutions, it is better to provide the user's knowledge about the problem as hints to improve its reasoning and problem-solving capabilities than a full solution for a similar problem, and hinting is the natural way in which humans solve a reasoning task as well, as only intermediate directions to approach a problem are required, the rest can be reasoned by humans themselves. Although extracting hints requires and extra inference, it is compensated by the increase in performance over other prompting techniques, which are only useful when dealing with similar problems. Finally, we see the sensitivity to random and adversarial prompting[18] techniques demonstrated performance loss due to adversarial/random hints in few-shot[3], one-shot[3], and chain-of-thought[4] settings.

# 5 Limitations and Future Work

Our work mainly focuses on prompting LLMs with problems as textual input. Testing these techniques in multi-modal models like VLMs, where geometrical problems can be passed as image inputs, is a possible future direction. Further, due to computational limitations, our experiments involved a small subset of problems to test the models. These techniques can also be evaluated on larger sample sizes to increase generalizability. Additionally, It is yet to be tested whether hinting can increase performance in other general tasks as well.

1. Our techniques are yet to be tested on larger models like the Llama 3.1 70B, 450B[19], Falcon 180B[20], OPT 175B[21], etc., and other closed source models like Claude Sonnet[22], Bard[23], etc due to computational limitations.
2. We compare the generated answers with the solutions using LLMs, which can introduce a degree of error.
3. These techniques can be further evaluated on the entire MATH dataset and other datasets such as GSM-8k[24], etc. to ensure a more exhaustive analysis.

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

# A  Prompting

We compare 17 different prompting techniques, including base-prompting techniques such as few-shot, one-shot, and chain-of-thought, and their variants involving hinting and adversarial prompting techniques for rigorous and exhaustive evaluation. We group the approaches as follows:

## A.1  Base Prompting techniques

In our baseline approach, we provide the models with the target questions and then prompt them to solve them. For One-Shot[3] and Few-shot prompting[3], we provide the model with example problems from the same class as the target problem. Since the dataset is divided into difficulty levels, we provide one Level-3 problem for one-shot and 5 examples, one from each level for few-shot prompting. We only provide the final numerical answer without any explanation for these approaches. More details about the difficulty levels are given in B. For Chain-of-Thought prompting[4], the model is given the complete step-by-step solution of the example problem in context instead of the final numerical answer only as shown in (Figure 4)

Our final baseline prompting techniques involve:

- **Baseline:** Only target question given
- **One-shot:** (Table 6) One example question and final answer pair (Difficulty level: 3) and then the target question given
- **Few-shot:** (Table 5) Giving five example questions and final answer pairs, each corresponding to one of the 5 difficulty levels, and then the target question
- **Chain Of Thought:** (Table 6) Giving one example QnA pair with step-by-step-solution (Difficulty level: 3) and then the target question

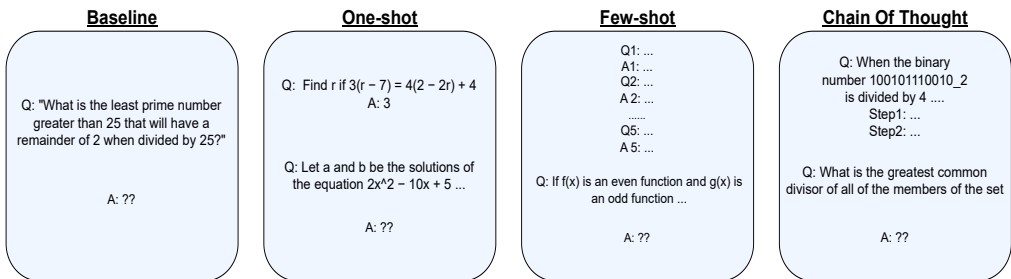

Figure 4: Examples of Base Prompting techniques

## A.2  Hinting

For our baseline approach, we only provide hints for the target problem. In the one-shot[3] and few-shot hinting cases[3], example problems of the same category and its hint (without the final answer) are provided to the model. (Figure 5 )

- **Baseline + Hinting:** Providing each question with its corresponding correct hint
- **One-shot-Hinting:** Providing one example Question and Hint pair (Difficulty level: 3) and then the target question
- **Few-shot-Hinting:** Providing five example Question and Hint pairs, each corresponding to one of the 5 difficulty levels, and then the target question

## A.3  Adversarial Prompts

To test the robustness of our approach, for one and few-shot, we adversarially prompt the model with an example problem and the incorrect final answer to test how much the models make use of the problem-answer pairs provided. We further extend the idea of adversarial prompting to chain of

Figure 5: Examples of Hint-based Prompting techniques: Hinting, One-shot Hinting, and Few-Shot Hinting

thought, by prompting the model with an example problem containing an erroneous step-by-step solution to the example problem.(Figure 6)

We experiment with two different types of adversarial prompting:

- **Case-I - Random :** The model is provided with an example problem of the same category as the target problem, but the hint being provided for the example problem is of a different category.

- **Case-II - Adversarial:** We deliberately make errors in the correct hints of the example problems and feed them to the model, such as changing the signs, reordering the steps, etc.

We experiment with both random and adversarial prompting for our baseline, one-shot, and few-shot approaches.

We further extend the concept of adversarial prompts to hinting and make adversarial hints to test the sensitivity of models to these hints. We add the adversarial hints case for one-shot and few-shot prompting cases similarly by replacing the right hints with the adversarial hints.

- **Baseline + Random-hint:** Giving a question with a random hint (hint of some unrelated question)

- **Baseline + Adversarial-hint:** Giving each question with its adversarial hint (wrong hint of the same question)

- **One-shot-Adversarial:** Giving one-example QnA pair but with the adversarial answer (Difficulty level: 3) and then the target question

- **One-shot-Random-Hinting:** Giving one example question and random hint (Case-I) pair (Difficulty level: 3) and then the target question

- **One-shot-Adversarial-Hinting:** Giving one example question and adversarial hint (Case-II) pair (Difficulty level: 3) and then the target question

- **Few-shot-Adversarial:** Giving five example QnA pairs with adversarial answers, each corresponding to one of the 5 difficulty levels, and then the target question

- **Few-shot-Random-Hinting:** Giving five example questions and random hints (Case-I) pair, each corresponding to one of the 5 difficulty levels, and then the target question

- **Few-shot-Adversarial-Hinting:** Giving five example questions and adversarial hints (Case-II) pair, each corresponding to one of the 5 difficulty levels, and then the target question

- **Chain-Of-Thought-Adversarial:** Giving one example QnA pair with the adversarial (Case-I) step-by-step-solution (Difficulty level: 3) and then the target question

- **Chain-Of-Thought-Random:** Giving one example QnA pair with random (Case-II) step-by-step-solution (Difficulty level: 3) and then the target question

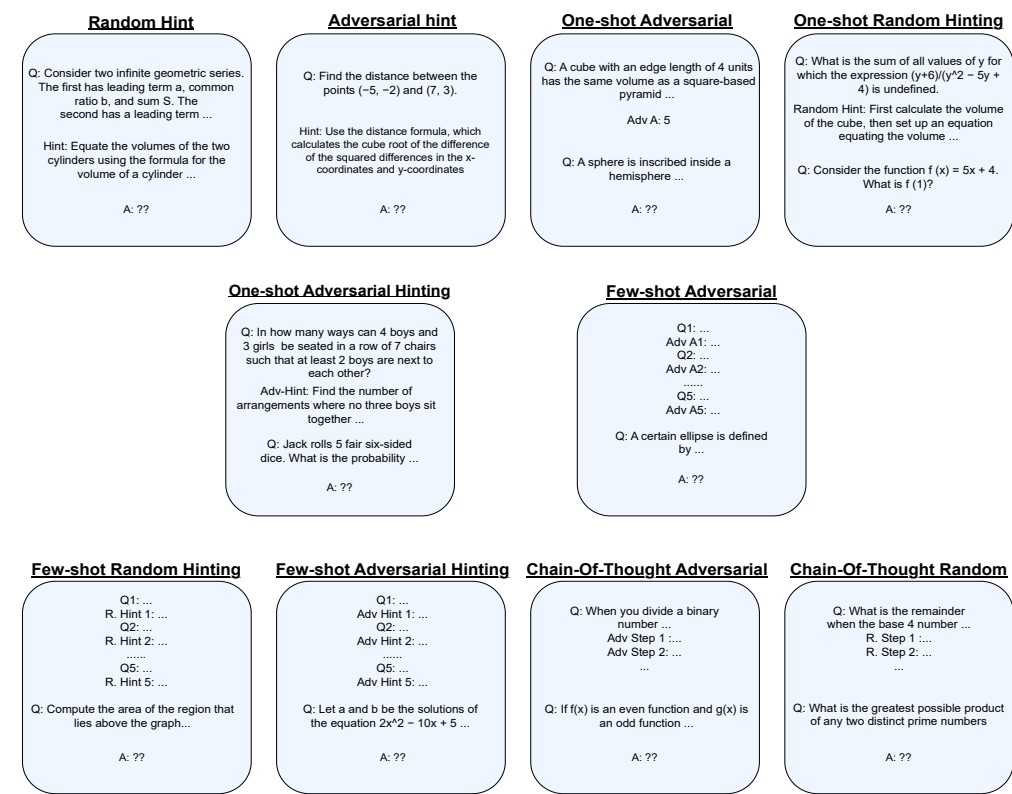

Figure 6: Examples of Adversarial Prompting techniques

# B Dataset

We evaluate on a subset of the MATH dataset [5]. The MATH dataset consists of problems from various popular math competitions, including the AMC 12, AIME, and more. Owing to the structure of mathematics, these problems have a particular method of solving them with multiple related steps. Humans generally use problem-solving techniques and "heuristics" to solve such problems, thus making them a good metric for assessing a model's problem-solving and reasoning skills[5]. The dataset categorizes the problems into various categories and difficulties. The seven categories are Pre-algebra, Algebra, Number Theory, Counting and Probability, Geometry, Intermediate Algebra, and Pre-calculus. The problems are divided into five levels: level 1, including more straightforward questions; and level 5, including advanced questions.

We enhanced the dataset with hints and adversarial hints for our experiments. To generate hints for each problem in the dataset, we used the Gemini-1.5-flash model, prompting it to generate hints and adversarial hints related to the question. Due to computational limitations, we used a subset of the dataset, 100 problems from each topic (20 questions of each difficulty), leading to a total sample set of 700 problems.

# C Experiments

All the prompting techniques were tested on a set of models, chosen in a manner to ensure diversity for our experimentation and testing. Nine open-sourced models and two closed-source have been used models as listed below:

- Gemma-2-2b-it[1]

---

[1]https://huggingface.co/google/gemma-2-2b-it

- Qwen2-Math-7B-Instruct[2]
- Meta-Llama-3.1-8B-Instruct[3]
- Meta-Llama-3.1-8B[4]
- Mistral-7B-Instruct-v0.3[5]
- Qwen2-7B-Instruct[6]
- Qwen2-7B[7]
- Mathstral-7B-v0.1[8]
- Deepseek-math-7b-instruct[9]
- Gemini
- GPT-4o mini

The ground truth numerical answers and the generated final numerical answers were compared using DeepSeek-7B-Math-Compare-Answer [25][10], fine-tuned to compare mathematical answers with high accuracy and extract answers from the generated step-by-step solution by the model.

### C.1 Experiments on VLMs

We evaluated the mathematical visual question answering ability of multi-modal models by providing them with the text for a question and an image containing the diagram. We tested the visual problem-solving with four prompting techniques: baseline, hinting, adversarial hinting, and random hinting. We evaluated these approaches on the multimodal Gemini and GPT-4o mini models. For our multimodal experiments, we created another dataset using a subset of 700 non-MCQ questions from the Math Vision Dataset[26], which contains problems of various levels from 1-5, in increasing difficulty. We sampled equally from each difficulty level. Hints were generated using the Gemini-1.5-Flash model using the textual content of the question. We also generated their adversarial hints in a similar manner as our other evaluations. For random hinting, we provided the hint from another question. As seen in Table 3, hinting improves the overall model performance, and the performance is degraded due to adversarial and random hinting, similar to our other evaluations. We note that we could not exhaustively evaluate a broader set of multimodal models on a larger dataset due to computational limitations and see it as an interesting future direction to explore.

| Model | Prompting method | Overall |
|-------|------------------|---------|
| **Gemini 1.5** | Baseline | 0.1714 |
| | Hinting | **0.2457** |
| | Adversarial Hinting | 0.1514 |
| | Random Hinting | 0.1657 |
| **GPT-4o mini** | Baseline | 0.1886 |
| | Hinting | **0.2514** |
| | Adversarial Hinting | 0.1886 |
| | Random Hinting | 0.1714 |

Table 3: Comparison of Visual Mathematical Question and Answering abilities of Multimodal models.

---

[2]https://huggingface.co/Qwen/Qwen2-Math-7B-Instruct

[3]https://huggingface.co/meta-llama/Meta-Llama-3.1-8B-Instruct

[4]https://huggingface.co/meta-llama/Meta-Llama-3.1-8B

[5]https://huggingface.co/mistralai/Mistral-7B-Instruct-v0.3

[6]https://huggingface.co/Qwen/Qwen2-7B-Instruct

[7]https://huggingface.co/Qwen/Qwen2-7B

[8]https://huggingface.co/mistralai/Mathstral-7B-v0.1

[9]https://huggingface.co/deepseek-ai/deepseek-math-7b-instruct

[10]https://huggingface.co/Tianqiao/DeepSeek-7B-Math-Compare-Answer

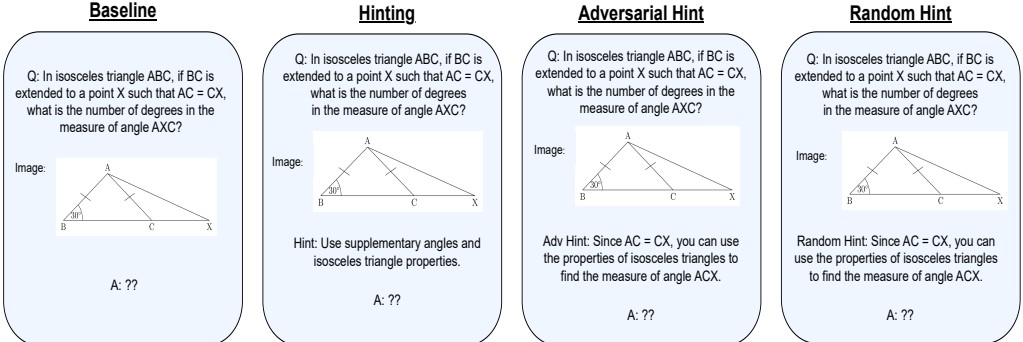

Figure 7: Examples of prompting techniques to test Visual problem solving ability

# D  Results

In this section, all the results from our experiments have been compiled to maintain transparency and clarity. Table 4 contains all the results for all the prompting techniques, individually mentioning the accuracy of each question type as well as model-wise performance. Figure 8 shows the average of all the models on a given technique, giving a better comparison of each type's performance.

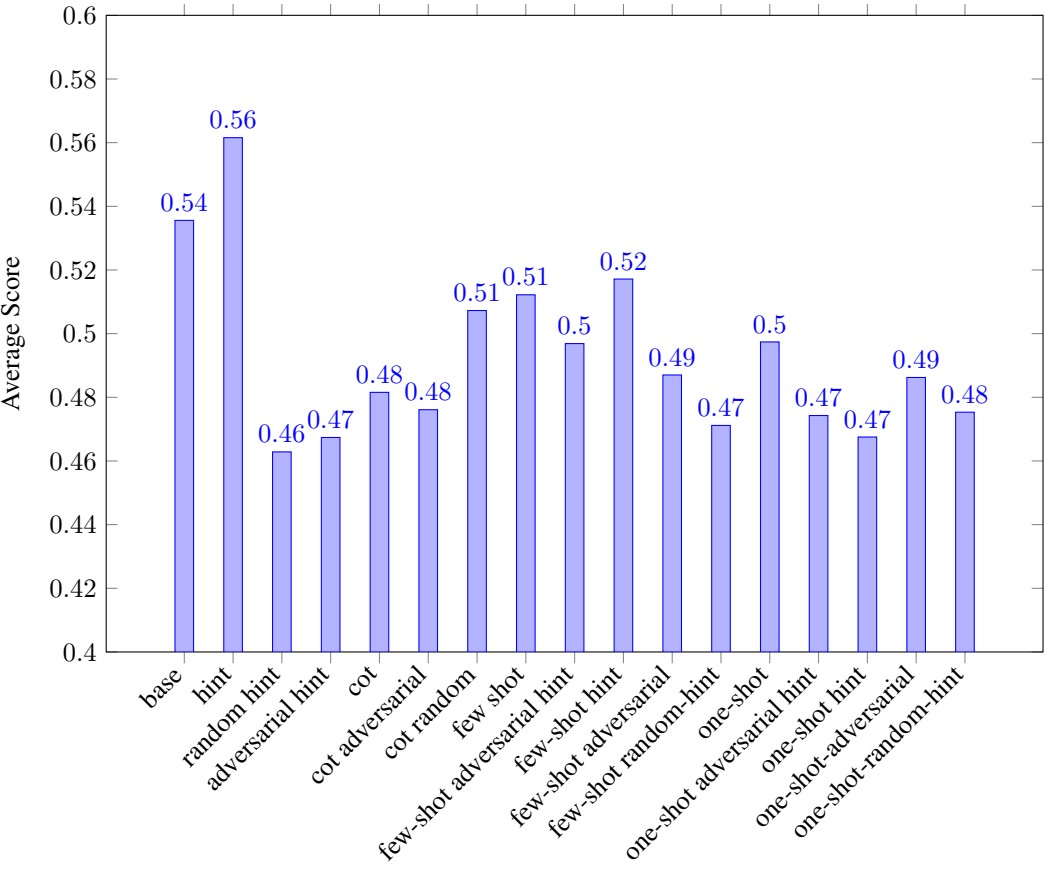

Figure 8:  Comparison of Average Scores for Different Methods

Table 4: Performance for all models for all prompting methods

| Model Name | Algebra | Count & Prob | Geometry | Int. Algebra | Number Theory | Pre-algebra | Pre-calculus | Overall |
|---|---|---|---|---|---|---|---|---|
| **Baseline** | | | | | | | | |
| Qwen2-Math-7B-Instruct | 0.86 | 0.82 | 0.68 | 0.68 | 0.85 | 0.9 | 0.58 | 0.7657 |
| Gemma-2-2b-it | 0.48 | 0.35 | 0.22 | 0.2 | 0.4 | 0.44 | 0.24 | 0.3314 |
| Meta-Llama-3.1-8B-Instruct | 0.72 | 0.44 | 0.42 | 0.58 | 0.46 | 0.76 | 0.64 | 0.5743 |
| Meta-Llama-3.1-8B | 0.56 | 0.4 | 0.62 | 0.59 | 0.41 | 0.62 | 0.7 | 0.5571 |
| Mistral-7B-Instruct-v0.3 | 0.18 | 0.21 | 0.11 | 0.22 | 0.12 | 0.4 | 0.16 | 0.1971 |
| Qwen2-7B-Instruct | 0.66 | 0.5 | 0.46 | 0.48 | 0.48 | 0.68 | 0.38 | 0.51 |
| Qwen2-7B | 0.6 | 0.54 | 0.58 | 0.34 | 0.46 | 0.68 | 0.44 | 0.52 |
| Mathstral-7B-v0.1 | 0.76 | 0.51 | 0.44 | 0.56 | 0.63 | 0.78 | 0.36 | 0.5771 |
| GPT-4o mini | 0.92 | 0.86 | 0.76 | 0.58 | 0.48 | 0.88 | 0.61 | 0.7257 |
| Deepseek-math-7b-instruct | 0.8 | 0.38 | 0.4 | 0.44 | 0.46 | 0.68 | 0.42 | 0.5114 |
| Gemini-1.5-Flash | 0.78 | 0.62 | 0.52 | 0.44 | 0.5 | 0.78 | 0.65 | 0.6114 |
| **Hint** | | | | | | | | |
| Qwen2-Math-7B-Instruct | 0.91 | 0.8 | 0.66 | 0.75 | 0.82 | 0.9 | 0.7 | 0.7914 |
| Gemma-2-2b-it | 0.46 | 0.28 | 0.3 | 0.36 | 0.48 | 0.5 | 0.2 | 0.3686 |
| Meta-Llama-3.1-8B-Instruct | 0.76 | 0.54 | 0.56 | 0.56 | 0.58 | 0.72 | 0.58 | 0.6143 |
| Meta-Llama-3.1-8B | 0.46 | 0.56 | 0.49 | 0.46 | 0.47 | 0.58 | 0.6 | 0.5143 |
| Mistral-7B-Instruct-v0.3 | 0.32 | 0.34 | 0.16 | 0.32 | 0.18 | 0.34 | 0.34 | 0.2857 |
| Qwen2-7B-Instruct | 0.58 | 0.42 | 0.52 | 0.42 | 0.62 | 0.58 | 0.4 | 0.5057 |
| Qwen2-7B | 0.54 | 0.56 | 0.52 | 0.42 | 0.52 | 0.5 | 0.48 | 0.5057 |
| GPT-4o mini | 0.96 | 0.88 | 0.78 | 0.52 | 0.56 | 0.94 | 0.74 | 0.7686 |
| Mathstral-7B-v0.1 | 0.69 | 0.5 | 0.51 | 0.49 | 0.68 | 0.72 | 0.55 | 0.5886 |
| Deepseek-math-7b-instruct | 0.76 | 0.52 | 0.56 | 0.46 | 0.56 | 0.58 | 0.52 | 0.5657 |
| Gemini-1.5-Flash | 0.78 | 0.78 | 0.58 | 0.54 | 0.52 | 0.86 | 0.62 | 0.6686 |
| **Adversarial-Hint** | | | | | | | | |
| Qwen2-Math-7B-Instruct | 0.86 | 0.72 | 0.62 | 0.74 | 0.82 | 0.80 | 0.62 | 0.7401 |
| Gemma-2-2B-it | 0.48 | 0.26 | 0.28 | 0.18 | 0.28 | 0.42 | 0.30 | 0.3143 |
| Meta-Llama-3.1-8B-Instruct | 0.70 | 0.44 | 0.52 | 0.46 | 0.42 | 0.58 | 0.44 | 0.5085 |
| Meta-Llama-3.1-8B | 0.46 | 0.36 | 0.46 | 0.60 | 0.32 | 0.56 | 0.65 | 0.4857 |
| Mistral-7B-Instruct-v0.3 | 0.20 | 0.18 | 0.10 | 0.28 | 0.16 | 0.24 | 0.16 | 0.1886 |
| Qwen2-7B-Instruct | 0.56 | 0.46 | 0.44 | 0.40 | 0.52 | 0.60 | 0.44 | 0.4886 |
| Qwen2-7B | 0.58 | 0.42 | 0.40 | 0.44 | 0.54 | 0.58 | 0.50 | 0.4942 |
| GPT-4o Mini | 0.92 | 0.90 | 0.76 | 0.50 | 0.56 | 0.92 | 0.68 | 0.7486 |
| Mathstral-7B-v0.1 | 0.60 | 0.46 | 0.34 | 0.50 | 0.60 | 0.66 | 0.36 | 0.5028 |
| Deepseek-Math-7B-Instruct | 0.70 | 0.38 | 0.36 | 0.28 | 0.46 | 0.54 | 0.30 | 0.4314 |
| Gemini-1.5-Flash | 0.82 | 0.70 | 0.44 | 0.34 | 0.44 | 0.78 | 0.42 | 0.5629 |
| **Random-Hint** | | | | | | | | |
| Qwen2-Math-7B-Instruct | 0.88 | 0.72 | 0.65 | 0.70 | 0.78 | 0.86 | 0.54 | 0.7314 |
| Gemma-2-2B-it | 0.38 | 0.14 | 0.20 | 0.16 | 0.22 | 0.42 | 0.10 | 0.2314 |
| Meta-Llama-3.1-8B-Instruct | 0.62 | 0.50 | 0.46 | 0.48 | 0.51 | 0.70 | 0.44 | 0.5314 |
| Meta-Llama-3.1-8B | 0.24 | 0.24 | 0.14 | 0.18 | 0.18 | 0.20 | 0.20 | 0.1971 |
| Mistral-7B-Instruct-v0.3 | 0.20 | 0.20 | 0.06 | 0.16 | 0.16 | 0.26 | 0.06 | 0.1571 |
| Qwen2-7B-Instruct | 0.62 | 0.44 | 0.34 | 0.42 | 0.50 | 0.58 | 0.34 | 0.4629 |
| Qwen2-7B | 0.38 | 0.32 | 0.34 | 0.40 | 0.30 | 0.54 | 0.36 | 0.3771 |
| Mathstral-7B-v0.1 | 0.68 | 0.48 | 0.48 | 0.50 | 0.64 | 0.72 | 0.32 | 0.5457 |
| Deepseek-Math-7B-Instruct | 0.66 | 0.38 | 0.40 | 0.32 | 0.58 | 0.62 | 0.34 | 0.4714 |
| Gemini-1.5-Flash | 0.88 | 0.66 | 0.60 | 0.42 | 0.50 | 0.84 | 0.56 | 0.6371 |
| GPT-4o Mini | 0.96 | 0.88 | 0.76 | 0.48 | 0.58 | 0.88 | 0.70 | 0.7486 |
| **Chain of Thought** | | | | | | | | |
| Qwen2-Math-7B-Instruct | 0.88 | 0.76 | 0.60 | 0.76 | 0.84 | 0.85 | 0.58 | 0.7514 |
| Gemma-2-2B-it | 0.42 | 0.26 | 0.24 | 0.26 | 0.32 | 0.44 | 0.18 | 0.3029 |
| Meta-Llama-3.1-8B-Instruct | 0.82 | 0.42 | 0.42 | 0.32 | 0.44 | 0.62 | 0.28 | 0.4743 |
| Meta-Llama-3.1-8B | 0.26 | 0.18 | 0.40 | 0.28 | 0.22 | 0.46 | 0.14 | 0.2771 |
| Mistral-7B-Instruct-v0.3 | 0.26 | 0.28 | 0.12 | 0.24 | 0.10 | 0.40 | 0.10 | 0.2143 |
| Qwen2-7B-Instruct | 0.26 | 0.36 | 0.32 | 0.36 | 0.22 | 0.34 | 0.24 | 0.3000 |
| Qwen2-7B | 0.60 | 0.54 | 0.58 | 0.48 | 0.58 | 0.52 | 0.54 | 0.5486 |
| Mathstral-7B-v0.1 | 0.72 | 0.52 | 0.48 | 0.52 | 0.62 | 0.70 | 0.28 | 0.5486 |
| Deepseek-Math-7B-Instruct | 0.78 | 0.46 | 0.42 | 0.40 | 0.48 | 0.66 | 0.30 | 0.5000 |
| Gemini-1.5-Flash | 0.78 | 0.72 | 0.62 | 0.48 | 0.52 | 0.84 | 0.56 | 0.6457 |
| GPT-4o Mini | 0.94 | 0.88 | 0.74 | 0.46 | 0.58 | 0.86 | 0.68 | 0.7343 |
| **CoT Adversarial** | | | | | | | | |
| Qwen2-Math-7B-Instruct | 0.88 | 0.72 | 0.62 | 0.74 | 0.88 | 0.88 | 0.62 | 0.7629 |
| Gemma-2-2B-it | 0.44 | 0.26 | 0.30 | 0.20 | 0.36 | 0.40 | 0.16 | 0.3029 |
| Meta-Llama-3.1-8B | 0.28 | 0.14 | 0.06 | 0.22 | 0.20 | 0.42 | 0.10 | 0.2029 |
| Mistral-7B-Instruct-v0.3 | 0.14 | 0.26 | 0.14 | 0.16 | 0.08 | 0.28 | 0.10 | 0.1657 |
| Qwen2-7B | 0.60 | 0.70 | 0.54 | 0.42 | 0.52 | 0.60 | 0.58 | 0.5657 |
| Meta-Llama-3.1-8B | 0.68 | 0.50 | 0.40 | 0.38 | 0.50 | 0.66 | 0.30 | 0.4886 |
| Qwen2-7B-Instruct | 0.24 | 0.28 | 0.22 | 0.36 | 0.22 | 0.36 | 0.28 | 0.2800 |
| Mathstral-7B-v0.1 | 0.78 | 0.54 | 0.54 | 0.46 | 0.60 | 0.74 | 0.30 | 0.5657 |
| Deepseek-Math-7B-Instruct | 0.70 | 0.44 | 0.40 | 0.38 | 0.48 | 0.66 | 0.42 | 0.4971 |
| Gemini-1.5-Flash | 0.90 | 0.72 | 0.56 | 0.40 | 0.56 | 0.82 | 0.56 | 0.6457 |
| GPT-4o Mini | 0.94 | 0.90 | 0.76 | 0.52 | 0.58 | 0.92 | 0.70 | 0.7600 |
| **CoT Random** | | | | | | | | |
| Qwen2-Math-7B-Instruct | 0.82 | 0.74 | 0.66 | 0.74 | 0.84 | 0.82 | 0.58 | 0.7429 |
| Gemma-2-2B-it | 0.58 | 0.24 | 0.32 | 0.28 | 0.32 | 0.38 | 0.18 | 0.3286 |
| Meta-Llama-3.1-8B | 0.58 | 0.30 | 0.54 | 0.76 | 0.34 | 0.32 | 0.28 | 0.4457 |
| Mistral-7B-Instruct-v0.3 | 0.16 | 0.18 | 0.08 | 0.24 | 0.16 | 0.42 | 0.16 | 0.2000 |
| Qwen2-7B | 0.52 | 0.48 | 0.56 | 0.48 | 0.50 | 0.70 | 0.44 | 0.5257 |
| Meta-Llama-3.1-8B | 0.74 | 0.46 | 0.38 | 0.48 | 0.60 | 0.65 | 0.38 | 0.5257 |
| Qwen2-7B-Instruct | 0.40 | 0.28 | 0.26 | 0.36 | 0.38 | 0.34 | 0.36 | 0.3400 |
| Mathstral-7B-v0.1 | 0.78 | 0.54 | 0.44 | 0.46 | 0.58 | 0.76 | 0.32 | 0.5543 |
| Deepseek-Math-7B-Instruct | 0.78 | 0.44 | 0.48 | 0.40 | 0.56 | 0.66 | 0.34 | 0.5129 |
| Gemini-1.5-Flash | 0.82 | 0.64 | 0.70 | 0.48 | 0.54 | 0.84 | 0.58 | 0.6571 |
| GPT-4o Mini | 0.96 | 0.86 | 0.78 | 0.52 | 0.52 | 0.86 | 0.66 | 0.7371 |

| Model Name | Algebra | Count & Prob | Geometry | Int. Algebra | Number Theory | Pre-algebra | Pre-calculus | Overall |
|---|---|---|---|---|---|---|---|---|
| **Few-Shot** | | | | | | | | |
| Qwen2-Math-7B-Instruct | 0.86 | 0.72 | 0.62 | 0.72 | 0.82 | 0.86 | 0.7 | 0.7571 |
| Gemma-2-2b-it | 0.48 | 0.26 | 0.14 | 0.26 | 0.32 | 0.42 | 0.14 | 0.2886 |
| Meta-Llama-3.1-8B-Instruct | 0.68 | 0.51 | 0.52 | 0.48 | 0.44 | 0.66 | 0.5 | 0.5429 |
| Meta-Llama-3.1-8B | 0.42 | 0.32 | 0.8 | 0.7 | 0.46 | 0.76 | 0.56 | 0.5743 |
| Mistral-7B-Instruct-v0.3 | 0.16 | 0.26 | 0.16 | 0.26 | 0.06 | 0.16 | 0.06 | 0.16 |
| Qwen2-7B-Instruct | 0.34 | 0.3 | 0.26 | 0.3 | 0.34 | 0.28 | 0.38 | 0.3143 |
| Qwen2-7B | 0.36 | 0.5 | 0.58 | 0.34 | 0.5 | 0.52 | 0.34 | 0.4486 |
| Mathstral-7B-v0.1 | 0.72 | 0.46 | 0.48 | 0.4 | 0.68 | 0.72 | 0.34 | 0.5429 |
| Deepseek-math-7b-instruct | 0.74 | 0.5 | 0.44 | 0.44 | 0.46 | 0.6 | 0.32 | 0.5 |
| Gemini-1.5-Flash | 0.8 | 0.84 | 0.74 | 0.74 | 0.7 | 0.8 | 0.7 | 0.76 |
| GPT-4o mini | 0.92 | 0.92 | 0.76 | 0.52 | 0.54 | 0.92 | 0.64 | 0.7457 |
| **Few-shot Hint** | | | | | | | | |
| Qwen2-Math-7B-Instruct | 0.78 | 0.74 | 0.56 | 0.76 | 0.82 | 0.85 | 0.58 | 0.7257 |
| Gemma-2-2b-it | 0.46 | 0.24 | 0.24 | 0.3 | 0.3 | 0.36 | 0.16 | 0.2943 |
| Meta-Llama-3.1-8B-Instruct | 0.72 | 0.6 | 0.5 | 0.44 | 0.52 | 0.7 | 0.4 | 0.5543 |
| Meta-Llama-3.1-8B | 0.64 | 0.44 | 0.96 | 1 | 0.6 | 0.8 | 0.92 | 0.7657 |
| Mistral-7B-Instruct-v0.3 | 0.22 | 0.24 | 0.18 | 0.18 | 0.14 | 0.32 | 0.2 | 0.2114 |
| Qwen2-7B-Instruct | 0.4 | 0.28 | 0.18 | 0.28 | 0.16 | 0.22 | 0.26 | 0.2543 |
| Qwen2-7B | 0.54 | 0.3 | 0.58 | 0.38 | 0.46 | 0.5 | 0.46 | 0.46 |
| Deepseek-math-7b-instruct | 0.74 | 0.48 | 0.4 | 0.36 | 0.46 | 0.68 | 0.44 | 0.5086 |
| Mathstral-7B-v0.1 | 0.64 | 0.44 | 0.4 | 0.38 | 0.6 | 0.7 | 0.36 | 0.5029 |
| Gemini-1.5-Flash | 0.74 | 0.72 | 0.68 | 0.58 | 0.52 | 0.76 | 0.5 | 0.6529 |
| GPT-4o mini | 0.9 | 0.88 | 0.88 | 0.5 | 0.54 | 0.92 | 0.74 | 0.7686 |
| **Few-Shot Adversarial** | | | | | | | | |
| Qwen2-Math-7B-Instruct | 0.82 | 0.80 | 0.54 | 0.68 | 0.84 | 0.82 | 0.62 | 0.73 |
| Gemma-2-2b-it | 0.50 | 0.28 | 0.18 | 0.20 | 0.36 | 0.38 | 0.12 | 0.29 |
| Meta-Llama-3.1-8B-Instruct | 0.60 | 0.48 | 0.50 | 0.52 | 0.52 | 0.70 | 0.46 | 0.54 |
| Meta-Llama-3.1-8B | 0.34 | 0.40 | 0.86 | 0.72 | 0.12 | 0.64 | 0.40 | 0.50 |
| Mistral-7B-Instruct-v0.3 | 0.14 | 0.22 | 0.08 | 0.24 | 0.16 | 0.22 | 0.10 | 0.17 |
| Qwen2-7B-Instruct | 0.52 | 0.26 | 0.34 | 0.26 | 0.28 | 0.30 | 0.28 | 0.32 |
| Qwen2-7B | 0.44 | 0.56 | 0.40 | 0.34 | 0.50 | 0.52 | 0.30 | 0.44 |
| Deepseek-Math-7B-Instruct | 0.76 | 0.54 | 0.40 | 0.40 | 0.52 | 0.68 | 0.32 | 0.51 |
| Mathstral-7B-v0.1 | 0.52 | 0.54 | 0.42 | 0.40 | 0.64 | 0.74 | 0.32 | 0.51 |
| Gemini-1.5-Flash | 0.76 | 0.62 | 0.60 | 0.42 | 0.42 | 0.82 | 0.42 | 0.58 |
| GPT-4o mini | 0.94 | 0.88 | 0.78 | 0.56 | 0.64 | 0.88 | 0.70 | 0.77 |
| **Few-Shot Adversarial Hint** | | | | | | | | |
| Qwen2-Math-7B-Instruct.json | 0.86 | 0.76 | 0.6 | 0.64 | 0.88 | 0.82 | 0.56 | 0.7314 |
| Gemma-2-2b-it.json | 0.42 | 0.32 | 0.22 | 0.2 | 0.34 | 0.38 | 0.08 | 0.28 |
| Mathstral-7B-v0.1.json | 0.72 | 0.5 | 0.36 | 0.5 | 0.62 | 0.78 | 0.44 | 0.56 |
| Meta-Llama-3.1-8B-Instruct.json | 0.72 | 0.48 | 0.54 | 0.48 | 0.58 | 0.74 | 0.42 | 0.5657 |
| Meta-Llama-3.1-8B.json | 0.22 | 0.2 | 0.66 | 0.74 | 0.22 | 0.8 | 0.36 | 0.4571 |
| Mistral-7B-Instruct-v0.3.json | 0.2 | 0.24 | 0.14 | 0.12 | 0.16 | 0.36 | 0.14 | 0.1943 |
| Qwen2-7B-Instruct.json | 0.32 | 0.3 | 0.14 | 0.2 | 0.2 | 0.32 | 0.24 | 0.2457 |
| Qwen2-7B.json | 0.56 | 0.56 | 0.62 | 0.46 | 0.6 | 0.58 | 0.52 | 0.5571 |
| Deepseek-math-7b-instruct.json | 0.65 | 0.48 | 0.44 | 0.44 | 0.56 | 0.66 | 0.4 | 0.5171 |
| Gemini-1.5-Flash | 0.82 | 0.66 | 0.6 | 0.54 | 0.46 | 0.78 | 0.52 | 0.6257 |
| GPT-4o mini | 0.92 | 0.82 | 0.8 | 0.54 | 0.48 | 0.88 | 0.68 | 0.7314 |
| **Few-Shot Random Hint** | | | | | | | | |
| Qwen2-Math-7B-Instruct | 0.90 | 0.66 | 0.58 | 0.74 | 0.74 | 0.82 | 0.54 | 0.71 |
| Mathstral-7B-v0.1 | 0.74 | 0.60 | 0.38 | 0.44 | 0.58 | 0.64 | 0.40 | 0.54 |
| Gemma-2-2b-it | 0.44 | 0.24 | 0.16 | 0.18 | 0.36 | 0.42 | 0.10 | 0.27 |
| Meta-Llama-3.1-8B-Instruct | 0.70 | 0.54 | 0.42 | 0.54 | 0.58 | 0.74 | 0.40 | 0.56 |
| Meta-Llama-3.1-8B | 0.24 | 0.16 | 0.62 | 0.44 | 0.46 | 0.76 | 0.56 | 0.46 |
| Mistral-7B-Instruct-v0.3 | 0.20 | 0.16 | 0.16 | 0.24 | 0.16 | 0.24 | 0.08 | 0.18 |
| Qwen2-7B-Instruct | 0.18 | 0.22 | 0.18 | 0.20 | 0.16 | 0.26 | 0.20 | 0.20 |
| Qwen2-7B | 0.26 | 0.44 | 0.36 | 0.54 | 0.34 | 0.38 | 0.38 | 0.39 |
| Deepseek-Math-7B-Instruct | 0.74 | 0.48 | 0.42 | 0.44 | 0.46 | 0.66 | 0.32 | 0.50 |
| Gemini-1.5-Flash | 0.78 | 0.72 | 0.56 | 0.46 | 0.50 | 0.80 | 0.52 | 0.62 |
| GPT-4o mini | 0.92 | 0.90 | 0.82 | 0.52 | 0.60 | 0.90 | 0.60 | 0.75 |
| **One-Shot** | | | | | | | | |
| Qwen2-Math-7B-Instruct | 0.80 | 0.72 | 0.56 | 0.72 | 0.74 | 0.86 | 0.40 | 0.69 |
| Mathstral-7B-v0.1 | 0.66 | 0.52 | 0.48 | 0.50 | 0.58 | 0.72 | 0.38 | 0.55 |
| Gemma-2-2b-it | 0.44 | 0.30 | 0.12 | 0.18 | 0.44 | 0.40 | 0.18 | 0.29 |
| Meta-Llama-3.1-8B-Instruct | 0.74 | 0.48 | 0.50 | 0.50 | 0.58 | 0.82 | 0.54 | 0.59 |
| Meta-Llama-3.1-8B | 0.48 | 0.24 | 0.60 | 0.40 | 0.24 | 0.30 | 0.34 | 0.37 |
| Mistral-7B-Instruct-v0.3 | 0.20 | 0.20 | 0.06 | 0.12 | 0.16 | 0.28 | 0.12 | 0.16 |
| Qwen2-7B-Instruct | 0.60 | 0.42 | 0.30 | 0.34 | 0.51 | 0.64 | 0.44 | 0.47 |
| Qwen2-7B | 0.52 | 0.56 | 0.42 | 0.50 | 0.52 | 0.52 | 0.44 | 0.50 |
| Deepseek-Math-7B-Instruct | 0.72 | 0.42 | 0.50 | 0.36 | 0.40 | 0.64 | 0.38 | 0.49 |
| Gemini-1.5-Flash | 0.74 | 0.68 | 0.74 | 0.42 | 0.52 | 0.74 | 0.54 | 0.63 |
| GPT-4o mini | 0.88 | 0.82 | 0.80 | 0.54 | 0.62 | 0.88 | 0.62 | 0.74 |
| **One-shot Hint** | | | | | | | | |
| Qwen2-Math-7B-Instruct | 0.80 | 0.80 | 0.68 | 0.68 | 0.80 | 0.80 | 0.52 | 0.73 |
| Mathstral-7B-v0.1 | 0.65 | 0.52 | 0.40 | 0.46 | 0.58 | 0.74 | 0.40 | 0.53 |
| Gemma-2-2b-it | 0.54 | 0.22 | 0.24 | 0.18 | 0.32 | 0.48 | 0.14 | 0.30 |
| Deepseek-Math-7B-Instruct | 0.68 | 0.42 | 0.42 | 0.46 | 0.44 | 0.60 | 0.36 | 0.48 |
| Meta-Llama-3.1-8B-Instruct | 0.78 | 0.54 | 0.44 | 0.56 | 0.54 | 0.66 | 0.34 | 0.55 |
| Meta-Llama-3.1-8B | 0.14 | 0.10 | 0.12 | 0.18 | 0.04 | 0.42 | 0.14 | 0.16 |
| Mistral-7B-Instruct-v0.3 | 0.28 | 0.26 | 0.08 | 0.22 | 0.12 | 0.30 | 0.18 | 0.21 |
| Qwen2-7B-Instruct | 0.48 | 0.38 | 0.32 | 0.30 | 0.40 | 0.38 | 0.42 | 0.38 |
| Qwen2-7B | 0.42 | 0.48 | 0.52 | 0.46 | 0.38 | 0.54 | 0.32 | 0.45 |
| Gemini-1.5-Flash | 0.78 | 0.62 | 0.70 | 0.44 | 0.48 | 0.80 | 0.46 | 0.61 |
| GPT-4o mini | 0.96 | 0.86 | 0.74 | 0.48 | 0.54 | 0.90 | 0.68 | 0.74 |

| Model Name | Algebra | Count & Prob | Geometry | Int. Algebra | Number Theory | Pre-algebra | Pre-calculus | Overall |
|---|---|---|---|---|---|---|---|---|
| **One-Shot Adversarial** | | | | | | | | |
| Qwen2-Math-7B-Instruct | 0.76 | 0.80 | 0.62 | 0.66 | 0.85 | 0.84 | 0.48 | 0.71 |
| Mathstral-7B-v0.1 | 0.74 | 0.50 | 0.46 | 0.50 | 0.51 | 0.82 | 0.42 | 0.57 |
| Gemma-2-2b-it | 0.46 | 0.28 | 0.20 | 0.26 | 0.30 | 0.38 | 0.16 | 0.29 |
| Meta-Llama-3.1-8B-Instruct | 0.70 | 0.58 | 0.44 | 0.58 | 0.42 | 0.84 | 0.44 | 0.57 |
| Meta-Llama-3.1-8B | 0.36 | 0.46 | 0.18 | 0.26 | 0.30 | 0.26 | 0.32 | 0.31 |
| Mistral-7B-Instruct-v0.3 | 0.24 | 0.16 | 0.12 | 0.16 | 0.12 | 0.32 | 0.14 | 0.18 |
| Qwen2-7B-Instruct | 0.56 | 0.44 | 0.46 | 0.48 | 0.38 | 0.60 | 0.36 | 0.47 |
| Qwen2-7B | 0.54 | 0.44 | 0.50 | 0.28 | 0.40 | 0.46 | 0.28 | 0.41 |
| Deepseek-Math-7B-Instruct | 0.66 | 0.48 | 0.44 | 0.34 | 0.44 | 0.70 | 0.32 | 0.48 |
| Gemini-1.5-Flash | 0.76 | 0.70 | 0.64 | 0.40 | 0.48 | 0.78 | 0.51 | 0.61 |
| GPT-4o mini | 0.94 | 0.88 | 0.78 | 0.50 | 0.54 | 0.90 | 0.66 | 0.74 |
| **One-Shot Adversarial Hint** | | | | | | | | |
| Qwen2-Math-7B-Instruct | 0.78 | 0.78 | 0.70 | 0.70 | 0.82 | 0.82 | 0.44 | 0.72 |
| Mathstral-7B-v0.1 | 0.54 | 0.52 | 0.42 | 0.52 | 0.64 | 0.78 | 0.36 | 0.54 |
| Deepseek-Math-7B-Instruct | 0.70 | 0.56 | 0.40 | 0.38 | 0.48 | 0.60 | 0.38 | 0.50 |
| Gemma-2-2b-it | 0.44 | 0.22 | 0.22 | 0.14 | 0.32 | 0.42 | 0.22 | 0.28 |
| Meta-Llama-3.1-8B-Instruct | 0.70 | 0.64 | 0.34 | 0.56 | 0.51 | 0.74 | 0.32 | 0.55 |
| Meta-Llama-3.1-8B | 0.04 | 0.20 | 0.18 | 0.12 | 0.02 | 0.12 | 0.32 | 0.14 |
| Mistral-7B-Instruct-v0.3 | 0.18 | 0.28 | 0.10 | 0.28 | 0.22 | 0.32 | 0.14 | 0.22 |
| Qwen2-7B-Instruct | 0.40 | 0.40 | 0.36 | 0.44 | 0.46 | 0.48 | 0.30 | 0.41 |
| Qwen2-7B | 0.54 | 0.51 | 0.51 | 0.36 | 0.46 | 0.60 | 0.42 | 0.49 |
| Gemini-1.5-Flash | 0.78 | 0.66 | 0.76 | 0.42 | 0.42 | 0.80 | 0.48 | 0.62 |
| GPT-4o mini | 0.94 | 0.88 | 0.82 | 0.54 | 0.56 | 0.88 | 0.68 | 0.76 |
| **One-Shot Random Hint** | | | | | | | | |
| Qwen2-Math-7B-Instruct | 0.88 | 0.78 | 0.60 | 0.60 | 0.82 | 0.82 | 0.44 | 0.71 |
| Mathstral-7B-v0.1 | 0.68 | 0.62 | 0.46 | 0.46 | 0.64 | 0.78 | 0.36 | 0.57 |
| Gemma-2-2b-it | 0.51 | 0.24 | 0.18 | 0.22 | 0.30 | 0.42 | 0.20 | 0.30 |
| Meta-Llama-3.1-8B-Instruct | 0.60 | 0.58 | 0.52 | 0.54 | 0.52 | 0.74 | 0.32 | 0.55 |
| Meta-Llama-3.1-8B | 0.06 | 0.20 | 0.18 | 0.22 | 0.02 | 0.12 | 0.32 | 0.16 |
| Mistral-7B-Instruct-v0.3 | 0.22 | 0.26 | 0.10 | 0.14 | 0.22 | 0.32 | 0.14 | 0.20 |
| Qwen2-7B-Instruct | 0.51 | 0.34 | 0.48 | 0.48 | 0.46 | 0.48 | 0.30 | 0.44 |
| Qwen2-7B | 0.44 | 0.44 | 0.56 | 0.38 | 0.46 | 0.60 | 0.42 | 0.47 |
| Deepseek-Math-7B-Instruct | 0.70 | 0.36 | 0.40 | 0.48 | 0.48 | 0.60 | 0.38 | 0.49 |
| Gemini-1.5-Flash | 0.72 | 0.70 | 0.62 | 0.36 | 0.50 | 0.86 | 0.50 | 0.61 |
| GPT-4o mini | 0.98 | 0.90 | 0.74 | 0.54 | 0.48 | 0.92 | 0.66 | 0.75 |

# E  Tables

Table 4: Few-Shot Prompting Examples for All Math Categories used for our experimentation

| Category | Few shot |
|---|---|
| **Algebra** | 1. **Example Problem**: "Let $$f(x) = \begin{cases} ax + 3, & \text{if } x > 2, \\ x - 5 & \text{if } -2 \leq x \leq 2, \\ 2x - b & \text{if } x < -2. \end{cases}$$ Find $a + b$ if the piecewise function is continuous (which means that its graph can be drawn without lifting your pencil from the paper)." **Answer**: 0 

 2. **Example Problem**: "Sixteen is $64\%$ of what number?" **Answer**: 25 

 3. **Example Problem**: "Karl was attempting to calculate economic figures. He found the following equation to be true: $$fp - w = 10000$$ If $f = 5$ and $w = 5 + 125i$, what is $p$?" **Answer**: $2001 + 25i$ 

 4. **Example Problem**: "What is the sum of all values of $y$ for which the expression $\frac{y+6}{y^2-5y+4}$ is undefined?" **Answer**: 5 

 5. **Example Problem**: "Find the distance between the points $(-5, -2)$ and $(7, 3)$." |

| Category | Few shot |
|---|---|
| | **Answer**: 13 |
| **Counting and Probability** | 1. **Example Problem**: "What is the value of $9^3 + 3(9^2) + 3(9) + 1$?"
**Answer**: 1000

2. **Example Problem**: "What is the coefficient of $x^8$ in the expansion of $(x-1)^9$?"
**Answer**: -9

3. **Example Problem**: "The Smith family has 4 sons and 3 daughters. In how many ways can they be seated in a row of 7 chairs such that at least 2 boys are next to each other?"
**Answer**: 4896

4. **Example Problem**: "John draws a regular five pointed star in the sand, and at each of the 5 outward-pointing points and 5 inward-pointing points he places one of ten different sea shells. How many ways can he place the shells, if reflections and rotations of an arrangement are considered equivalent?"
**Answer**: 362880

5. **Example Problem**: "Compute $\binom{17}{9}$. You are told that $\binom{15}{6} = 5005$ and $\binom{15}{8} = 6435$."
**Answer**: 24310 |
| **Geometry** | 1. **Example Problem**: "Square ABCD has its center at $(8, -8)$ and has an area of 4 square units. The top side of the square is horizontal. The square is then dilated with the dilation center at (0,0) and a scale factor of two. What are the coordinates of the vertex of the image of square ABCD that is farthest from the origin? Give your answer as an ordered pair."
**Answer**: (18,-18)

2. **Example Problem**: "In triangle $ABC$, we have that $E$ and $F$ are midpoints of sides $\overline{AC}$ and $\overline{AB}$, respectively. The area of $\triangle ABC$ is 24 square units. How many square units are in the area of $\triangle CEF$?"
**Answer**: 6

3. **Example Problem**: "The consecutive angles of a particular trapezoid form an arithmetic sequence. If the largest angle measures $120°$, what is the measure of the smallest angle?"
**Answer**: $60°$

4. **Example Problem**: "Triangle $ABC$ with vertices $A(-2, 0)$, $B(1, 4)$ and $C(-3, 2)$ is reflected over the $y$-axis to form triangle $A'B'C'$. What is the length of a segment drawn from $C$ to $C'$?"
**Answer**: 6

5. **Example Problem**: "A cube with an edge length of 4 units has the same volume as a square-based pyramid with base edge lengths of 8 units and a height of $h$ units. What is the value of $h$?"
**Answer**: 3 |

| Category | Few shot |
| --- | --- |
| **Intermediate Algebra** | 1. **Example Problem**: "Let $a_1, a_2, \ldots$ be a sequence for which $a_1 = 2$, $a_2 = 3$, and $a_n = \frac{a_{n-1}}{a_{n-2}}$ for each positive integer $n \geq 3$. What is $a_{2006}$?" 
 **Answer**: 3 |

**Intermediate Algebra**

1. **Example Problem**: "Let $a_1, a_2, \ldots$ be a sequence for which $a_1 = 2$, $a_2 = 3$, and $a_n = \frac{a_{n-1}}{a_{n-2}}$ for each positive integer $n \geq 3$. What is $a_{2006}$?"
**Answer**: 3

2. **Example Problem**: "When a polynomial is divided by $2x^2 - 7x + 18$, what are the possible degrees of the remainder? Enter all the possible values, separated by commas."
**Answer**: 0,1

3. **Example Problem**: "Karl was attempting to calculate economic figures. He found the following equation to be true:

$$fp - w = 10000$$

If $f = 5$ and $w = 5 + 125i$, what is $p$?""Let $a$ and $b$ be nonzero real numbers such that
$$(2 - 7i)(a + bi)$$
is pure imaginary. Find $\frac{a}{b}$."
**Answer**: -7/2

4. **Example Problem**: "Find all real solutions to $x^4 + (2 - x)^4 = 34$. Enter all the solutions, separated by commas."
**Answer**: $1 + \sqrt{2}$, $1 - \sqrt{2}$

5. **Example Problem**: Shown below are rows 1, 2, and 3 of Pascal's triangle.

$$
\begin{array}{ccccccc}
 & & 1 & & 1 & & \\
 & 1 & & 2 & & 1 & \\
1 & & 3 & & 3 & & 1
\end{array}
$$

Let $(a_i)$, $(b_i)$, $(c_i)$ be the sequence, from left to right, of elements in the 2005th, 2006th, and 2007th rows, respectively, with the leftmost element occurring at $i = 0$. Compute

$$\sum_{i=0}^{2006} \frac{b_i}{c_i} - \sum_{i=0}^{2005} \frac{a_i}{b_i}.$$

**Answer**: 1/2

**Number theory**

1. **Example Problem**: "If $AAA_4$ can be expressed as $33_b$, where $A$ is a digit in base 4 and $b$ is a base greater than 5, what is the smallest possible sum $A + b$?"
**Answer**: 7

2. **Example Problem**: "Abigail, Beatrice, and Carson combine their eggs to sell them at the market. If Abigail has 37 eggs, Beatrice has 49 eggs, and Carson has 14 eggs, and if eggs can only be sold in cartons of 12, how many eggs will be left over if all cartons are sold?"
**Answer**: 4

3. **Example Problem**: "For how many positive integers $n$ does $\frac{1}{n}$ yield a terminating decimal with a non-zero hundredths digit?"
**Answer**: 11

| Category | Few shot |
|---|---|
| | 4. **Example Problem**: "Find the remainder when the sum |

$$75 + 76 + 77 + 78 + 79 + 80 + 81 + 82$$

is divided by 16."
**Answer**: 4

5. **Example Problem**: "When the binary number $100101110010_2$ is divided by 4, what is the remainder (give your answer in base 10)?"
**Answer**: 2

**Pre-algebra**

1. **Example Problem**: "Find the area in square feet of a square with a perimeter of 32ft."
**Answer**: 64

2. **Example Problem**: "Find $r$ if $3(r - 7) = 4(2 - 2r) + 4$."
**Answer**: 3

3. **Example Problem**: "Megan has lost Fatima's phone number. Megan knows that the first three digits are either 296 or 299. The remaining four digits are 0, 1, 6 and 7, but she isn't sure of the order of these digits. If Megan randomly dials a seven-digit number that meets these conditions, what is the probability that she dials Fatima's correct number? Express your answer as a common fraction."
**Answer**: 1/48

4. **Example Problem**: "What is $\frac{2}{5}$ divided by 3?"
**Answer**: 2/15

5. **Example Problem**: "Twenty gremlins and fifteen imps are at the Annual Mischief Convention. The imps have had a lot of in-fighting lately and refuse to shake hands with each other, but they readily shake hands with all of the gremlins. Meanwhile, all the gremlins are quite friendly and shake hands with all of the other gremlins as well as imps. Each pair of creatures shakes hands at most once. How many handshakes were at the convention?"
**Answer**: 490

**Pre-calculus**

1. **Example Problem**: Compute

$$\begin{vmatrix} 7 & 3 \\ -1 & 2 \end{vmatrix}.$$

**Answer**: 17

2. **Example Problem**: "Let

$$\mathbf{A} = \begin{pmatrix} 0 & 1 & 2 \\ 1 & 0 & 1 \\ 2 & 1 & 0 \end{pmatrix}.$$

There exist constants $p$, $q$, and $r$ such that

$$\mathbf{A}^3 + p\mathbf{A}^2 + q\mathbf{A} + r\mathbf{I} = \mathbf{0},$$

where $\mathbf{I}$ and $\mathbf{0}$ are the $3 \times 3$ identity matrix and zero matrix, respectively. Enter the ordered triple $(p, q, r)$."
**Answer**: (0,-6,-4)

| Category | Few shot |
| --- | --- |

3. **Example Problem**: "Let $S$ be a region in the plane with area 10. When we apply the matrix

$$\begin{pmatrix} 2 & 1 \\ 7 & -3 \end{pmatrix}$$

to $S$, we obtain the region $S'$. Find the area of $S'$."
**Answer**: 130

4. **Example Problem**: "Find the least positive integer $n$ such that

$$\frac{1}{\sin 45° \sin 46°} + \frac{1}{\sin 47° \sin 48°} + \cdots + \frac{1}{\sin 133° \sin 134°} = \frac{1}{\sin n°}.$$

**Answer**: 1

5. **Example Problem**: "Find the area of the triangle with vertices $(-1, 4)$, $(7, 0)$, and $(11, 5)$."
**Answer**: 28

Table 5: Chain-Of-Thought Prompting Examples for All Math Categories used for experimentation

| Category | COT |
| --- | --- |
| **Algebra** | **Example Problem**: "What is the sum of all values of $y$ for which the expression $\frac{y+6}{y^2-5y+4}$ is undefined?"
**Solution**:
* Step 1: Understanding the Problem: The expression is undefined when its denominator is zero. We need to find the values of $y$ that make the denominator zero.
* Step 2: Identify the Denominator: The denominator is $y^2 - 5y + 4$. We need to find values of $y$ for which this quadratic expression equals zero.
* Step 3: Set the Denominator to Zero: Set the quadratic equation to zero: $y^2 - 5y + 4 = 0$.
* Step 4: Finding the Sum of the Zeros: The sum of the solutions (or zeros) of a quadratic equation $ax^2 + bx + c = 0$ is given by $-\frac{b}{a}$.
* Step 5: Apply the Formula: Here, $a = 1$, $b = -5$, and $c = 4$. Using the formula: Sum of the zeros $= -\frac{-5}{1} = 5$.
* Step 6: Conclusion: The sum of the zeros of $y^2 - 5y + 4$ is $\boxed{5}$. |
| **Counting and Probability** | **Example Problem**: "What is the coefficient of $x^8$ in the expansion of $(x - 1)^9$?"

**Solution**:
* Step 1: Understanding the Problem: Find the coefficient of a specific term in the expansion of $(x - 1)^9$ using the Binomial Theorem. |

| Category | COT |
|---|---|
| | * Step 2: Recall the Binomial Theorem: For $(a+b)^n$, the expansion is: $$(a+b)^n = \sum_{k=0}^{n} \binom{n}{k} a^{n-k} b^k$$ where $\binom{n}{k}$ is the binomial coefficient. 
 * Step 3: Identify the Term: We're interested in the term where $x^8$ appears in the expansion of $(x-1)^9$. In the general term, $a = x$ and $b = -1$. 
 * Step 4: Apply the Binomial Theorem: The general term is given by: $$\binom{9}{k} x^{9-k} (-1)^k$$ For $x^8$, $9 - k = 8$, so $k = 1$. 
 * Step 5: Calculate the Coefficient: Substitute $k = 1$ to find the coefficient: $$\binom{9}{1} x^8 (-1)^1 = 9 \times (-1) = -9$$ * Step 6: Conclusion: The coefficient of $x^8$ in the expansion of $(x-1)^9$ is $\boxed{-9}$. |
| **Geometry** | **Example Problem**: "A cube with an edge length of 4 units has the same volume as a square-based pyramid with base edge lengths of 8 units and a height of $h$ units. What is the value of $h$?" 

 **Solution**: 
 * Step 1: Determine the Volume of the Cube: The volume of the cube is given by $4^3$. We calculate this as: $$4^3 = 64$$ So, the volume of the cube is 64 cubic units. 
 * Step 2: Write the Volume Formula for the Pyramid: The volume $V$ of a pyramid is given by: $$V = \frac{1}{3} \times \text{base area} \times \text{height}$$ For this pyramid, the base area is $8^2$. Thus, the volume formula becomes: $$V = \frac{1}{3} \times 8^2 \times h$$ . 
 * Step 3: Set Up the Equation: We know the volume of the pyramid is equal to the volume of the cube, which is 64. So, we set up the equation: $$64 = \frac{1}{3} \times 8^2 \times h$$ . 
 * Step 4: Simplify the Equation: Calculate $8^2$: $$8^2 = 64$$ |

| Category | COT |
|---|---|
| | Substitute this into the volume formula: |

$$64 = \frac{64}{3} \times h$$

.

* Step 5: Solve for $h$: Rearrange the equation to solve for $h$:

$$64 = \frac{64}{3} \times h$$

To isolate $h$, multiply both sides by 3:

$$64 \times 3 = 64 \times h$$

. Divide both sides by 64:

$$h = \frac{64 \times 3}{64} = 3$$

* Step 6: Conclusion: Therefore, the height $h$ of the pyramid is $\boxed{3}$.

---

**Intermediate Algebra**

**Example Problem**: "Let $a$ and $b$ be nonzero real numbers such that $(2 - 7i)(a + bi)$ is pure imaginary. Find $\frac{a}{b}$."
**Solution**:
* Step 1: Expand the Expression: Expand the given expression using the distributive property:

$$(2 - 7i)(a + bi) = 2(a + bi) - 7i(a + bi)$$
$$= 2a + 2bi - 7ai - 7bi^2$$

* Step 2: Simplify Using $i^2 = -1$: Recall that $i^2 = -1$, so $-7bi^2 = 7b$. Substituting this gives:

$$= 2a + 2bi - 7ai + 7b$$

* Step 3: Combine Like Terms: Group the real and imaginary parts:

$$= (2a + 7b) + (-7a + 2b)i$$

* Step 4: Identify the Pure Imaginary Condition: The expression is pure imaginary, meaning the real part must be zero. Set the real part equal to zero:
$$2a + 7b = 0$$

* Step 5: Solve for $\frac{a}{b}$: Rearrange the equation to solve for $a$:
$$2a = -7b$$

Divide by $b$:

$$\frac{a}{b} = -\frac{7}{2}$$

* Step 6: Conclusion: Therefore, the value of $\frac{a}{b}$ is $\boxed{-\frac{7}{2}}$.

| Category | COT |
| --- | --- |
| **Number Theory** | **Example Problem**: "When the binary number $100101110010_2$ is divided by 4, what is the remainder (give your answer in base 10)?" |

**Solution**:
* Step 1: Understand the Binary Number Representation: The binary number $b_k b_{k-1} \ldots b_2 b_1 b_0$ represents a decimal number where each digit $b_i$ is multiplied by $2^i$. The decimal value of the binary number is given by:
$$2^k b_k + 2^{k-1} b_{k-1} + \cdots + 4b_2 + 2b_1 + b_0$$

* Step 2: Find the Remainder When Dividing by 4: To find the remainder when this number is divided by 4, observe that only the last two binary digits affect the remainder. This is because $2^2 = 4$ is the base of the modulus. Therefore:
$$\text{Remainder} = 2b_1 + b_0$$

* Step 3: Apply the Method to the Given Binary Number: Consider the binary number $100101110010_2$. Identify the last two digits of this binary number:
$$\text{Last two digits} = 10$$

* Step 4: Calculate the Remainder: Use the formula $2b_1 + b_0$ with $b_1 = 1$ and $b_0 = 0$:
$$\text{Remainder} = 2 \cdot 1 + 0 = 2$$

* Step 5: Conclusion: Therefore, when the binary number $100101110010_2$ is divided by 4, the remainder is $\boxed{2}$.

---

**Pre-algebra**

**Example Problem**: "Find $r$ if $3(r - 7) = 4(2 - 2r) + 4$"

**Solution**:
* Step 1: Expand Both Products: We start with the equation involving products. Expanding both products, we get:
$$3r - 3 \cdot 7 = 4 \cdot 2 - 4 \cdot 2r + 4$$

* Step 2: Calculate the Products: Compute each product separately:
$$3r - 21 = 8 - 8r + 4$$

* Step 3: Simplify the Right-Hand Side: Combine the terms on the right-hand side:
$$8 - 8r + 4 = 12 - 8r$$
Thus, the equation becomes:
$$3r - 21 = 12 - 8r$$

| Category | COT |
|---|---|
| | * Step 4: Solve for $r$: To isolate $r$, first add $8r$ to both sides of the equation:

$$3r - 21 + 8r = 12 - 8r + 8r$$

This simplifies to:

$$11r - 21 = 12$$

* Step 5: Add 21 to Both Sides: To isolate $11r$, add 21 to both sides:

$$11r - 21 + 21 = 12 + 21$$

This simplifies to:

$$11r = 33$$

* Step 6: Solve for $r$: Divide both sides by 11 to find $r$:

$$r = \frac{33}{11} = 3$$

* Step 7: Conclusion: Therefore, the value of $r$ is $\boxed{3}$. |
| **Pre-calculus** | **Example Problem**: "If $\cos x + \cos 2x + \cos 3x = 3$, then find $\sin x + \sin 2x + \sin 3x$."

**Solution**:

• Step 1: Understand the Given Expression: We are given that $\cos x + \cos 2x + \cos 3x = 3$.

• Step 2: Identify Possible Values for the Cosine Function: Recall that the maximum value of $\cos \theta$ is 1. Therefore, for $\cos x + \cos 2x + \cos 3x = 3$, each term must be at its maximum value, which means $\cos x = \cos 2x = \cos 3x = 1$.

• Step 3: Solve for $x$: For $\cos 2x = 1$, substitute $x = 2n\pi$ into $\cos 2x$, giving $\cos 4n\pi = 1$, which is true. For $\cos 3x = 1$, substitute $x = 2n\pi$ into $\cos 3x$, giving $\cos 6n\pi = 1$, which is true. Thus, $x = 2n\pi$ satisfies all conditions.

• Step 4: Calculate $\sin x + \sin 2x + \sin 3x$: For $x = 2n\pi$, we have $\sin x = \sin 2n\pi = 0$. Similarly, $\sin 2x = \sin 4n\pi = 0$. And $\sin 3x = \sin 6n\pi = 0$.

• Step 5: Conclusion: Adding these results, we get $\sin x + \sin 2x + \sin 3x = 0 + 0 + 0 = 0$. Therefore, the value of $\sin x + \sin 2x + \sin 3x$ is $\boxed{0}$. |

Table 6: One-Shot Prompting Examples for All Math Categories used for our experimentation

| Category | One shot |
|---|---|
| **Algebra** | **Example Problem**: "What is the sum of all values of $y$ for which the expression $\frac{y+6}{y^2-5y+4}$ is undefined?" 

 **Answer**: 5 |
| **Counting and Probability** | **Example Problem**: "What is the coefficient of $x^8$ in the expansion of $(x-1)^9$?" 

 **Answer**: -9 |
| **Geometry** | **Example Problem**: "A cube with an edge length of 4 units has the same volume as a square-based pyramid with base edge lengths of 8 units and a height of $h$ units. What is the value of $h$?" 

 **Answer**: 3 |
| **Intermediate algebra** | **Example Problem**: "Let $a$ and $b$ be nonzero real numbers such that $$(2-7i)(a+bi)$$ is pure imaginary. Find $\frac{a}{b}$." 

 **Answer**: 7/2 |
| **Number theory** | **Example Problem**: "When the binary number $100101110010_2$ is divided by 4, what is the remainder (give your answer in base 10)?" 

 **Answer**: 2 |
| **Pre-algebra** | **Example Problem**: "Find $r$ if $3(r-7) = 4(2-2r)+4$." 

 **Answer**: 3 |
| **Pre-calculus** | **Example Problem**: "If $\cos x + \cos 2x + \cos 3x = 3$, then find $\sin x + \sin 2x + \sin 3x$." 

 **Answer**: 0 |

