# OpenReview forum: "Give me a hint: Can LLMs take a hint to solve math problems?"
_NeurIPS.cc/2024/Workshop/MATH-AI — MATH-AI 24_

### Official Review · Reviewer_Lz1P · 2024-10-06
**A good paper but have some questions**

**Rating:** 7
**Confidence:** 4

**Review:**

### Summary

The paper explores the use of "hints" as a new prompting technique to improve the problem-solving capabilities of LLMs mathematical tasks. The study is inspired by pedagogical approaches in human math learning, where hints are used to guide learners towards a solution without giving away the full answer. The authors compare this hinting method against other prompting methods such as one-shot, few-shot, and chain of thought (CoT) prompting using various LLMs across the MATH dataset.

### Strengths

1. Introducing hinting as a way to guide models through the reasoning process seems like a new approach. It mimics human pedagogy, where learners are given clues to nudge them in the right direction. I am slightly curious to know how this hinting method fundamentally changes the way in which the LLMs give outputs. Some insights from the authors would be good to have.
2. The paper evaluates multiple LLMs, including fine-tuned, instruction-tuned, and base models. This diversity helps demonstrate the general applicability of the hinting technique across different types of models, and provides a well-rounded comparison of model performances.
3. Testing the robustness of models against adversarial hints is a good addition. It provides insights into how sensitive LLMs are to misleading information.

### Weaknesses

1. **Limited Dataset Scope**: The experiments are conducted on a subset of the MATH dataset. While the results are promising, testing on a more extensive range of datasets like GSM-8k or broader math domains would provide a more thorough validation of the hinting technique.
2. **Adversarial and Random Hinting Unexplored in Depth**: While the paper mentions the drop in performance due to adversarial and random hinting, it lacks a deep exploration of the failure modes and reasons why certain models are more sensitive to these hints. A more granular error analysis could strengthen the paper’s conclusions.

### Questions

1. How would the hinting technique perform on larger-scale models, such as GPT-4 or Llama 3.1, which might already exhibit stronger mathematical reasoning capabilities?
2. Could the hinting method be extended to non-mathematical domains (e.g., language understanding or scientific reasoning) where guidance is also essential for problem-solving?
3. What specific types of problems in the MATH dataset benefited most from hinting versus other techniques like CoT or few-shot prompting? A deeper dive into problem categories might help identify where hinting is most advantageous.
4. Would the performance drop due to adversarial hints be more pronounced in models fine-tuned for specific domains? Can future work offer solutions to mitigate these adversarial effects?

---

### Official Review · Reviewer_RwaB · 2024-10-08
**The authors propose giving hints to improve mathematical problem-solving in large language models (LLMs). Hinting is compared to existing prompting methods on a range of math problems from the MATH dataset, with experiments on various open-source and closed-source LLMs. The results show hinting generally outperforms baselines like chain-of-thought prompting.**

**Rating:** 6
**Confidence:** 4

**Review:**

The paper proposes giving "hints" to large language models (LLMs) as a technique to improve their mathematical problem-solving performance. The hints are intended to guide the models to the correct solution in a manner analogous to how hints are used pedagogically with humans. The authors compare the hinting approach to other prompting techniques like one-shot, few-shot, and chain-of-thought prompting on a diverse set of math problems from the MATH dataset. They also test the models' robustness to misleading adversarial hints. The experiments are conducted on a range of open-source and closed-source LLMs.


Pros:
- The paper addresses an important and timely topic of improving the mathematical reasoning capabilities of LLMs, which is critical for many potential applications.
- The hinting technique proposed is intuitive and well-motivated by drawing an analogy to human math pedagogy. Providing hints to guide the model's reasoning process is a natural extension of existing prompting techniques.
- The experimental evaluation is thorough, covering a diverse set of math problems of varying types and difficulties from the MATH dataset. This helps establish the generality of the findings.
- Comparing hinting to strong baseline techniques like one-shot, few-shot, and chain-of-thought prompting helps contextualize the results and demonstrate its advantages.
- Testing robustness to adversarial hints is a valuable addition that explores the hinting approach's limitations and failure modes.


Cons:
- The hint generation process could be described in more detail. How exactly were the hints extracted from the Gemini model? Were they manually curated or filtered in any way?
- The authors mention that hinting outperforms CoT prompting and attribute this to CoT overfitting to the example solution and getting derailed by mistakes. However, this could be discussed and analyzed further. Are there some problems where CoT still performs better than hinting?
- The adversarial hinting experiments are useful but could be expanded. How exactly were the adversarial hints generated? Are they mathematically invalid or just suboptimal? Varying the types of adversarial hints could yield further insights.

---

### Official Review · Reviewer_gXha · 2024-10-08
**Prompting method seems effective but lacks novelty**

**Rating:** 5
**Confidence:** 4

**Review:**

This work demonstrates that providing hints before generating solutions can improve model performance. The method proves effective across multiple models when compared to baseline approaches like Chain of Thought (COT) and few-shot learning. While the experiments are thorough and the results seem promising, the approach of introducing hints is neither novel nor particularly impressive, and can be regarded as a special case of COT. Therefore, I would rate this work a 5.

---

### Official Review · Reviewer_zsxN · 2024-10-08
**A Promising Approach to Enhancing LLM Mathematical Reasoning**

**Rating:** 5
**Confidence:** 5

**Review:**

This paper investigates the use of "hinting" to enhance LLMs' performance on mathematical tasks, comparing it with other prompting strategies and assessing robustness to adversarial hints. Using a variety of models and problems from the MATH dataset, the study shows that hinting generally leads to better performance than baseline, one-shot, few-shot, and chain-of-thought prompting. While providing valuable insights into LLM math capabilities and prompting techniques, the work has some limitations in depth and scope.

The paper follows a clear structure but could be improved in several areas:
1. The abstract should be more concise and highlight the key findings.
2. The methodology section is dense and could be organized better with subheadings.
3. Figure captions need to be more descriptive for better data interpretation.
4. A more thorough literature review would help contextualize the research.

Methodologically, the paper is strong, though certain aspects could be refined:
1. The experimental setup is comprehensive, covering multiple models and problem types.
2. The comparison between hinting and other prompting techniques is well-executed.
3. The use of the MATH dataset provides a standardized benchmark.
4. Adversarial robustness testing is a valuable addition.
5. However, the statistical analysis could be improved with clearer reporting of significance and effect sizes.

Strengths of the paper :
1. Innovative approach to LLM math performance through hinting.
2. Extensive evaluation across models and problem types.
3. Inclusion of adversarial testing for robustness.
4. Clear visualizations of results.
5. Practical applications for AI in math problem-solving.

Weaknesses of the paper:
1. Limited sample size may reduce generalizability.
2. Lacks in-depth analysis of why hinting outperforms other techniques.
3. No comparison to human performance or advanced math-solving systems.
4. Insufficient exploration of optimal hint generation processes.
5. Potential biases or shortcuts introduced by hinting are not fully addressed.

Comments and Questions :
1. How does hint quality affect results? A deeper analysis would be helpful.
2. What are the implications for AI-driven math education tools or tutoring systems?
3. Could hinting be combined with chain-of-thought reasoning for even better performance?
4. A more detailed error analysis would help understand persistent model failures.
5. How does performance vary across different difficulty levels within each math category?
6. What are the computational costs of hinting compared to other methods, especially for larger models?
7. Can this approach generalize beyond mathematics?

This paper offers contributions to understanding how LLMs handle mathematical reasoning. The hinting technique shows promise, although the study has some limitations. Addressing these gaps and further exploring the practical implications would strengthen future work.

---

### Official Review · Reviewer_19dt · 2024-10-09

**Rating:** 6
**Confidence:** 4

**Review:**

Strengths:
* Potential for a more comprehensive analysis of different prompting techniques for mathematical reasoning, with a focus on hinting and effects of adversarial hinting

Weaknesses:
* The idea of hinting is not novel
* Limited evaluations (also see questions below)

Questions:
* What is the effect of various techniques in prompt engineering? The examples in figure 6 seem to suggest that little effort was put into prompt engineering.
* What are the results with zero-shot CoT? They usually improve over the baseline.
* How were the in-context learning examples chosen? What do the results look like with smarter retrieval for in-context learning examples?

Suggestions:
* Appendix D, with the full evaluation results, should be referred to in the main text.

---

### Decision · Program_Chairs · 2024-10-09

Accept